# Comparison of seven SNP calling pipelines for the next-generation sequencing data of chickens

**Jing Liu, Qingmiao Shen, Haigang Bao** *

National Engineering Laboratory for Animal Breeding, Beijing Key Laboratory for Animal Genetic Improvement, College of Animal Science and Technology, China Agricultural University, Beijing, China

* zjbhg@126.com

## Abstract

Single nucleotide polymorphisms (SNPs) are widely used in genome-wide association studies and population genetics analyses. Next-generation sequencing (NGS) has become convenient, and many SNP-calling pipelines have been developed for human NGS data. We took advantage of a gap knowledge in selecting the appropriated SNP calling pipeline to handle with high-throughput NGS data. To fill this gap, we studied and compared seven SNP calling pipelines, which include 16GT, genome analysis toolkit (GATK), Bcftools-single (Bcftools single sample mode), Bcftools-multiple (Bcftools multiple sample mode), VarScan2-single (VarScan2 single sample mode), VarScan2-multiple (VarScan2 multiple sample mode) and Freebayes pipelines, using 96 NGS data with the different depth gradients of approximately 5X, 10X, 20X, 30X, 40X, and 50X coverage from 16 Rhode Island Red chickens. The sixteen chickens were also genotyped with a 50K SNP array, and the sensitivity and specificity of each pipeline were assessed by comparison to the results of SNP arrays. For each pipeline, except Freebayes, the number of detected SNPs increased as the input read depth increased. In comparison with other pipelines, 16GT, followed by Bcftools-multiple, obtained the most SNPs when the input coverage exceeded 10X, and Bcftools-multiple obtained the most when the input was 5X and 10X. The sensitivity and specificity of each pipeline increased with increasing input. Bcftools-multiple had the highest sensitivity numerically when the input ranged from 5X to 30X, and 16GT showed the highest sensitivity when the input was 40X and 50X. Bcftools-multiple also had the highest specificity, followed by GATK, at almost all input levels. For most calling pipelines, there were no obvious changes in SNP numbers, sensitivities or specificities beyond 20X. In conclusion, (1) if only SNPs were detected, the sequencing depth did not need to exceed 20X; (2) the Bcftools-multiple may be the best choice for detecting SNPs from chicken NGS data, but for a single sample or sequencing depth greater than 20X, 16GT was recommended. Our findings provide a reference for researchers to select suitable pipelines to obtain SNPs from the NGS data of chickens or nonhuman animals.

**Data Availability Statement:** The DNA sequencing and genotyping data for this study can be downloaded from the China National GeneBank (Accession numbers: CNP0001419 and CNP0001435).

**Funding:** This study was supported by the Modern Agricultural Industry Technology System of China [grant number CARS-40]. The funder did not play any role in the design of the study, collection, analysis, interpretation of data or writing the manuscript.

**Competing interests:** The authors have declared that no competing interests exist.

**Abbreviations:** Bcftools-multiple, Bcftools multiple sample mode; Bcftools-single, Bcftools single sample mode; FP, the false genotype with false positive SNPs; GATK, genome analysis toolkit; GE, the false genotype with true positive SNPs; MG, the missing genotypes from sequencing data at the positive array sites; NGS, next-generation sequencing; SNP, single nucleotide polymorphisms; Ti/Tv, transition/transversion ratio; TP, the true genotype with true positive SNPs; VarScan2-multiple, VarScan2 multiple sample mode; VarScan2-single, VarScan2 single sample mode.

# Introduction

In the last decade, next-generation sequencing (NGS) has been extensively used in human, livestock and plant research [1–5]. An increasing number of single nucleotide polymorphisms (SNPs) have been detected in NGS datasets using various calling pipelines [6–8]. SNPs might occur at nonspecific positions in the genome and have been widely used in genome-wide association studies and population genetics analyses [9]. Many SNPs related to complex diseases or traits in humans or animals have been discovered by whole-genome sequencing and whole-exome sequencing [10]. Some SNPs have been shown to be causal mutations of some traits or diseases [11,12].

Many variant calling pipelines have been developed to detect SNPs from NGS data; however, each pipeline has its own advantages and disadvantages [13]. The genome analysis toolkit (GATK, https://software.broadinstitute.org/gatk/) [14] and Bcftools (https://samtools.github.io/bcftools/bcftools.html) [15] may be the most widely used SNP calling pipelines to date. A brief characteristic summary of several calling tools is listed in Table 1 and described as follows. GATK was originally used to analyze human genome and exome sequencing data, and now it may be regarded as the industry standard for identifying SNPs in germline DNA and RNA NGS data [14]. The toolkit contains a wide variety of tools with a primary focus on variant discovery and genotyping. Bcftools is a high-speed program for calling variants. It can manipulate variant calls in compressed/uncompressed VCF and BCF files [15]. VarScan2 (http://varscan.sourceforge.net/using-varscan.html) is the first tool used for the detection of somatic mutations and copy number alterations in exome data from tumor-normal pairs [16]. The VarScan2 algorithm reads the SAMtools pileup or mpileup output of tumor and normal samples simultaneously, performs pairwise comparisons of base calls, and normalizes sequencing depths at each position [17]. Freebayes (https://github.com/ekg/freebayes) is a Bayesian genetic variant caller designed to find SNPs, indels, multinucleotide polymorphisms, and complex events (composite insertion and substitution events) smaller than the length of a short-read sequencing alignment [18]. Freebayes uses short-read alignments for any number of individuals from a population and uses a reference genome to determine the most likely combination of genotypes at each position in the population [18]. 16GT (https://github.com/aquaskyline/16GT) is the first publicly available caller that uses a 16-genotype probabilistic model to unify SNPs and indel calling in a single algorithm [19]. Compared with the traditional 10-genotype probabilistic model, 16GT added 6 new genotypes. Compared to GATK with HaplotypeCaller, 16GT not only runs 4 times faster but also improves sensitivity in calling SNPs by unifying SNPs and indel calling in a single algorithm of variant calling. Recently, Chiara et al. also provided a consensus variant calling system, CoVaCS (https://bioinformatics.cineca.it/covacs), for the analysis of human genome resequencing studies [20].

**Table 1. A brief summary of different tools.**

| caller | Bcftools | 16GT | Freebayes | VarScan2 | GATK |
|--------|----------|------|-----------|----------|------|
| Code | C | Perl | C++ | Java | Java |
| Model | HMM & MAQ | 16-genotype probabilistic | Bayesian | heuristic algorithm | Bayesian |
| Sampling | Single & multiple | Single | Single | Single & multiple | Single & multiple |
| Variants | SNPs & indels | SNPs & indels | SNPs & indels&MNPs | SNPs & indels | SNPs & indels |
| Features | Sorting, indexing, etc. | easy to use, timesaving | straightforward | meet desired thresholds for read depth, base quality, variant allele frequency, and statistical significance | Realignment, per base recalibration, VQSR |
| Reference | Danecek et al., 2017 [15] | Luo et al., 2017 [19] | Garrison and Marth, 2012 [18] | Koboldt et al., 2012 [16] | Mckenna et al., 2010 [14] |

Using simulation and real NGS data of humans, many studies have shown that different tools have their own advantages and disadvantages [6,8,12,21]. Different variant callers may produce different results, so ensemble methods of variant calling algorithms or analytic pipelines can improve variant accuracy [22,23]. However, a single pipeline, such as the pipelines of BWA-MEM and GATK-HaplotypeCaller, can be run similarly to the pipeline ensemble method [23]. GATK may be the most popular pipeline for detecting SNPs from human high-throughput data sets [24], and it has also been widely used in chicken NGS data in recent studies [25–27]. Compared with known human variant information resources, the corresponding resources of chickens are quite few, which may affect the detection results if we use GATK to detect SNPs from chicken data. Ni et al. [7] compared variants detected with GATK (Unified-Genotyper and hard filtering), Freebayes, and SAMtools using chicken NGS data with an average coverage of 7.6 X and found that all three pipelines, particularly GATK and SAMtools, perform well in general. In the present study, we used NGS data from 16 Rhode Island Red chickens to evaluate seven SNP calling pipelines, including 16GT, GATK, Bcftools-single (Bcftools single sample mode), Bcftools-multiple (Bcftools multiple sample mode), VarScan2-single (VarScan2 single sample mode), VarScan2-multiple (VarScan2 multiple sample mode), and Freebayes, in terms of the number of detected SNPs, sensitivity, and specificity. We aim to select a high-performance SNP calling pipeline for chicken NGS data studies.

## Materials and methods

### Ethics statement

All experimental procedures and animals used were approved by the Ethics Review Committee for Laboratory Animal Welfare and Animal Experiment of China Agricultural University (Approval number: AW70101202-1-1).

### Animals and DNA samples

The animal experimental process complied with the regulations and guidelines of the Experimental Animal Welfare and Animal Experiment Ethics Review Committee of China Agricultural University. A total of 16 chickens at 18 weeks of age randomly selected from the Rhode Island Red population, and blood samples were collected from each chicken's wing vein using 2 mL injectors. After blood was collected, we put the 16 chickens back to the population and keep them with other individuals reared in the Experimental Chicken Farm of China Agricultural University. Our subsequent research did not work with animals. Genomic DNA of blood was extracted using the TIANamp Genomic DNA Kit (Cat. #DP304-02, TIANGEN) according to the protocol supplied. After checking and qualification, each DNA sample was divided into two parts, one part for next-generation sequencing (paired-end sequencing, 150 bp, 50X, Illumina HiSeq™ 4000, Beijing Novogene Bioinformatics Technology Co., Ltd) and the other for SNP array analyses (50K, KPS CAULayer Breeding Chip v1, Beijing Compass Biotechnology Co., Ltd, S1 Table).

### NGS data sets and SNP calling pipelines

Cleaned reads were obtained by Trimmomatic (version 0.39; S1 Word) from raw sequencing data. After quality control, the cleaned data of each of the 16 samples were split into 10 parts evenly and reorganized to form 6 subsets of various sequencing depth gradients of approximately 5X, 10X, 20X, 30X, 40X, and 50X coverage according to Bentley et al. [28]. Thus, we finally had 16 samples × 6 gradients = 96 data points. Bowtie 2 [29] was chosen as the common aligner with the chicken genome reference (Gallus_gallus-5.0) for all SNP calling pipelines in

the present study. We conducted alignment with Bowtie 2, converted the SAM files to BAM files, and then processed the same BAM files with seven SNP calling pipelines, including 16GT, GATK, Bcftools-single, Bcftools-multiple, VarScan2-single, VarScan2-multiple and Freebayes. All results of this study depended on programs' defaults in each pipeline. Details of processing with all these pipelines are described in S1 Word.

## Analysis of the sensitivity and specificity of SNP-calling pipelines

We compared the SNP array genotypes with the genotypes of SNP loci in the array detected by sequencing pipelines. In order to assess the sensitivity, and specificity of the pipelines with input read depth gradients of 5X-50X coverage, SNP loci in the array that were also detected from sequencing data for each individual were divided into 4 categories (Table 2) referring to Liu et al. [6] as follows: (1) sequencing SNPs with matched array genotypes (the true genotype with true positive SNPs (TP)); (2) false genotypes from sequencing data at the matched positive array sites (the false genotype with true positive SNPs (GE)); (3) false genotypes from sequencing data with negative array genotypes (the false genotype with false positive SNPs (FP)); and (4) the missing genotypes from sequencing data at the positive array sites (MG). Four metrics, including the SNP number, sensitivity, specificity and transition/transversion ratio (Ti/Tv), were used to assess the performance of each SNP calling pipeline. The SNP number indicates the number of detected SNPs in each sample at any input read depth. The sensitivity of each pipeline was calculated as (TP + GE)/(TP + GE + MG), and the specificity was calculated as TP/(TP + FP + GE). The Ti/Tv ratios were calculated using VCFtools (Version 0.1.17) [30].

## Statistical analysis

Means and standard errors were calculated for the SNP number, sensitivity and specificity of each pipeline at each input level. Mean differences were tested by the Duncan test of SPSS 19.0 (SPSS Inc., Chicago, IL), and the statistical significance level was set at *P < 0.05*.

## Results

### The NGS data sets and alignment

Approximately 3.5 billion paired-end cleaned data reads were obtained with an average coverage of approximately 50X for each sequenced Rhode Island Red chicken (S2 Table). The cleaned data set of each sample was split into 10 parts evenly and reorganized, and we obtained a total of 96 data sets. Each sample had 6 data sets with different coverages of approximately 5X, 10X, 20X, 30X, 40X and 50X (S3 Table). Paired-end cleaned reads were aligned against the chicken reference genome (Gallus_gallus-5.0) using Bowtie 2 (version 2.2.9). A summary of cleaned data alignments is displayed in S3 Table. The alignment rate of the cleaned data of each sample was between 90.91% and 95.21% (S3 Table).

**Table 2. Descriptions of genotype categories.**

| Genotype categories | | Genotype from SNP array | | |
|---|---|---|---|---|
| | | **00** | **01** | **11** |
| Genotype from sequencing data | 01 | FP | TP, MG | GE |
| | 11 | FP | GE | TP, MG |

*Notes: TP means sequencing SNPs with matched array genotypes (The true genotype with true positive SNPs); GE means false genotypes from sequencing data at the matched positive array sites (The false genotype with true positive SNPs); FP means false genotypes from sequencing data with negative array genotypes (the false genotype with false positive SNPs) and MG means the missing genotypes from sequencing data at the positive array sites.

## Comparisons of the numbers of SNPs detected by different SNP calling pipelines

The numbers of SNPs detected with different input read depths are shown in Fig 1 and S4 Table. From Fig 1B, we could see that an increasing number of SNPs were detected with increasing input read depths by each variant caller except Freebayes. When the sequencing depth was less than 20X, the number of SNPs found by any caller increased rapidly with increasing sequencing depth, while when the sequencing depth was greater than 20X, the speed of increase slowed down obviously, and Freebayes even reached the maximum at 20X (Fig 1B). In comparison with other callers, 16GT obtained the most abundant SNPs at almost all input read depths (except 5X) in the present study; VarScan2-single and VarScan2-multiple obtained the same SNP numbers at all input read depths, and both called out the fewest SNPs at low sequencing depths (< 20X), while Freebayes called the fewest SNPs at high sequencing depths (> = 20X), and GATK and Bcftools-single performed moderately (Fig 1A). From Fig 1A, we could also see that Bcftools-multiple obtained the most abundant SNPs at 5X and 10X input levels, and at high input depths (> = 20X), Bcftools-multiple also obtained higher SNP numbers in comparison with any other pipeline except 16GT.

## Comparisons of the sensitivity and specificity among the seven SNP calling pipelines

To assess the sensitivity, and specificity of each pipeline with different input read depths, a 50K chicken SNP array (KPS CAULayer Breeding Chip v1, Beijing Compass Biotechnology Co.,

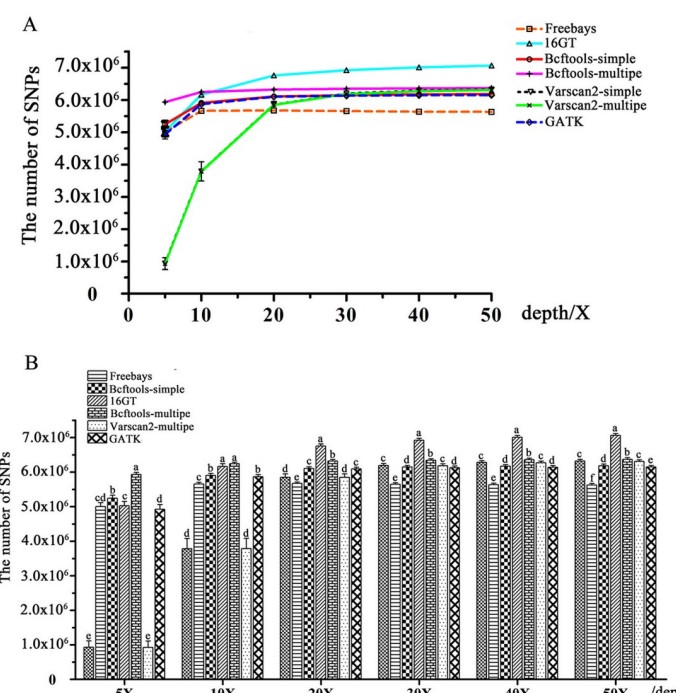

**Fig 1. Comparisons of the total number of SNPs called out by seven different SNP calling pipelines.** A: Comparisons of the number of SNPs called out by different calling pipelines at each input read depth level. For each input level, the same letters indicate that the difference is not significant ($P > 0.05$), and the different letters indicate significant differences ($P < = 0.05$). B: The tendency of the number of SNPs called out by each pipeline with increasing input level.

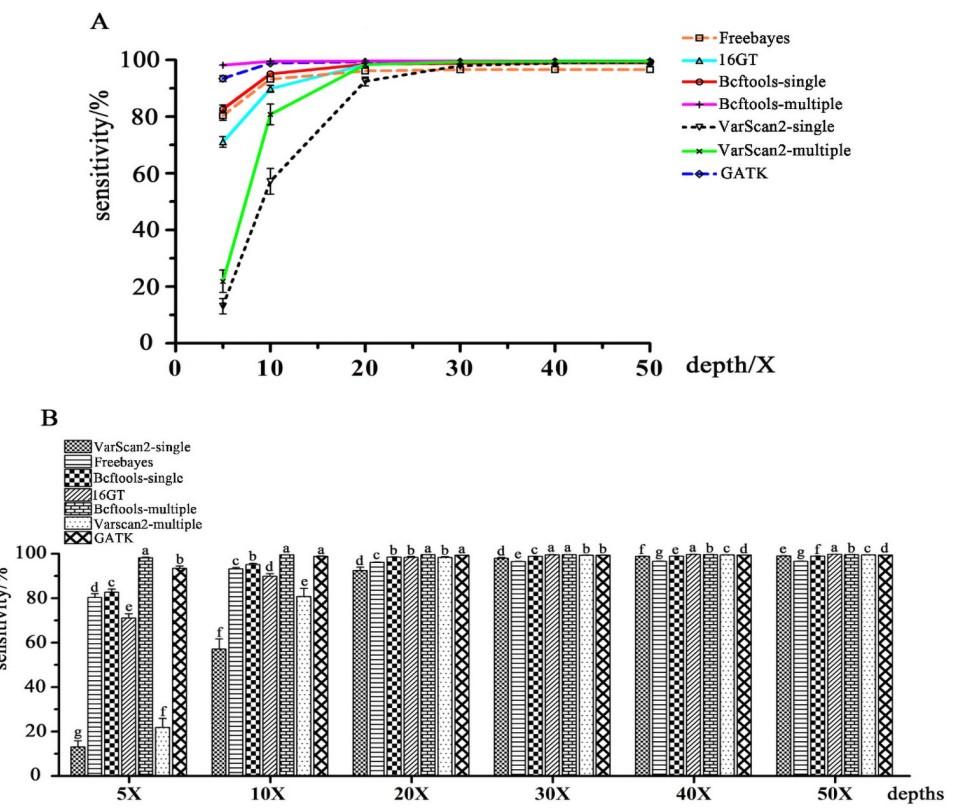

**Fig 2. The sensitivities of seven SNP calling pipelines.** A: The sensitivity tendencies of each SNP calling pipeline with the input level increasing; B: Comparisons of the sensitivities of different calling pipelines at each input read depth level. For each input level, the same letters indicate that the difference is not significant ($P > 0.05$), and different letters indicate significant differences ($P <= 0.05$).

Ltd, Beijing, China) with a total of 43,681 SNP sites (S1 Table) was used to genotype individuals. We compared the SNP array genotypes with the genotypes of SNP loci in the array detected by sequencing pipelines, and the array results were regarded as a standard to evaluate the specificity and sensitivity of each calling pipeline. The array results showed an average call rate of 99.20% (S5 Table).

The sensitivity of each pipeline is displayed in Figs 2 and 4 and S6 Table. As shown in Fig 2, the sensitivity of various pipelines tended to rapidly increase at lower input read depths and then slightly increase at higher input read depths with increasing sequencing depth. In comparison with any other pipeline in the present study, 16GT had higher sensitivity when input read depths were equal to or greater than 20X, and Freebayes showed its sensitivity moderately at lower sequencing depths ($<= 20X$) but the lowest from 30X to 50X. The two VarScan2 pipelines displayed the lowest sensitivity but increased rapidly at the low input read depths and then tended to stabilize. In Fig 2, Bcftools-multiple showed the best sensitivity from 5X to 30X input depths and was then exceeded by 16GT. GATK and Bcftools-multiple both showed the best sensitivity at 10X and 20X input depths, as shown in Fig 2B.

The differences in specificity among the seven pipelines were similar to the differences in sensitivity among them. Fig 3 and S7 Table show the specificities of the seven SNP calling pipelines at different input depths for SNP calling. From Fig 3, we observed that the specificity of each pipeline increased as the input read depth increased. In comparison with any other calling pipeline in the present study, Bcftools-multiple had higher specificity with any input read

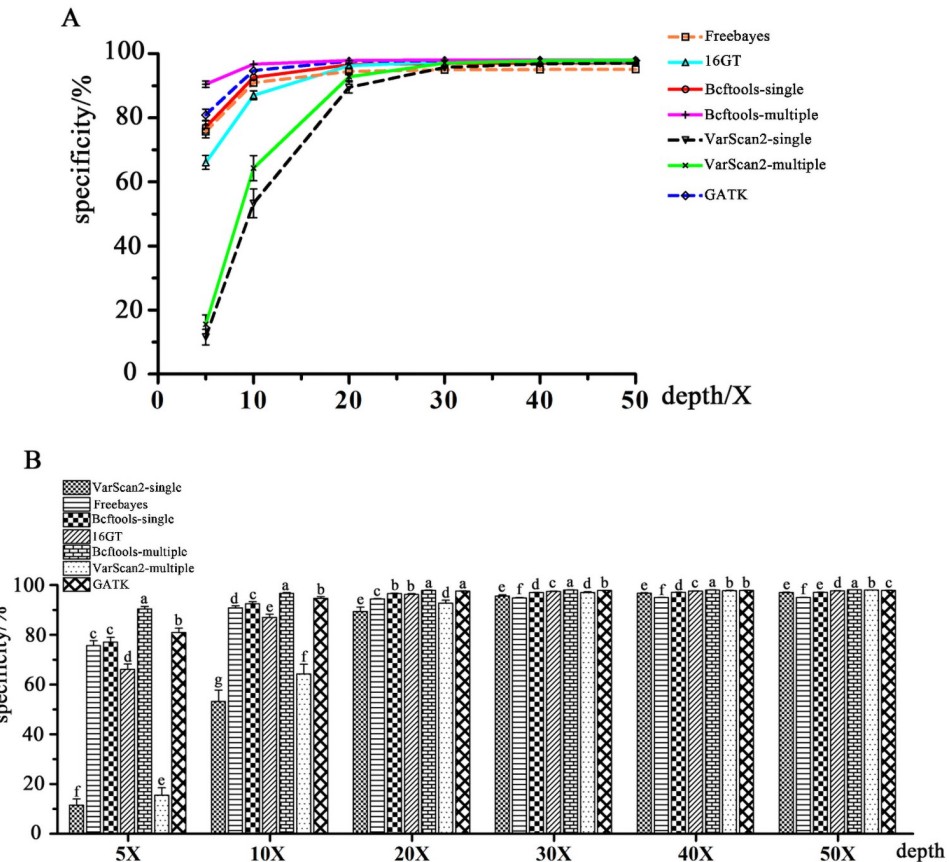

**Fig 3. The specificities of seven SNP calling pipelines.** A: The specificity tendencies of each SNP calling pipeline with the input level increasing; B: Comparisons of the specificities of different calling pipelines at each input read depth level. The same letter indicates that the difference is not significant ($P > 0.05$), and different letters indicate significant differences ($P < = 0.05$).

depth in the present study (Fig 3B). 16GT showed moderate specificity at any read depth. Compared with other pipelines, the two VarScan2 pipelines displayed the lowest specificity, but it increased rapidly at the low input read depths ($< = 20X$), while Freebayes showed the lowest specificity at the high input read depths ($> = 30X$). GATK had better specificity than any other pipeline at 5X to 40X input read depths except Bcftools-multiple in the present study.

Two-dimensional scatter plots with the specificities and sensitivities of seven SNP calling pipelines in different input read depths are displayed in Fig 4. From Fig 4, we can see that Bcftools-multiple may be the best pipeline in most cases considering both sensitivity and specificity.

## Effects of single and multiple modes on the sensitivity and specificity of Bcftools and VarScan2 Pipelines

Bcftools and VarScan2 can process files one by one (Bcftools-single and VarScan2-single pipelines) or multiple files once a time (Bcftools-multiple and VarScan2-multiple pipelines). From Fig 5, we could see that the sensitivity and specificity of calling procedures increased with increasing input read depth whether in a one-by-one way or multiple files a time. Bcftools-multiple and VarScan2-multiple had higher sensitivity and specificity than Bcftools-single and VarScan2-single, respectively (Fig 5; S6 and S7 Tables). Especially at low input read depths,

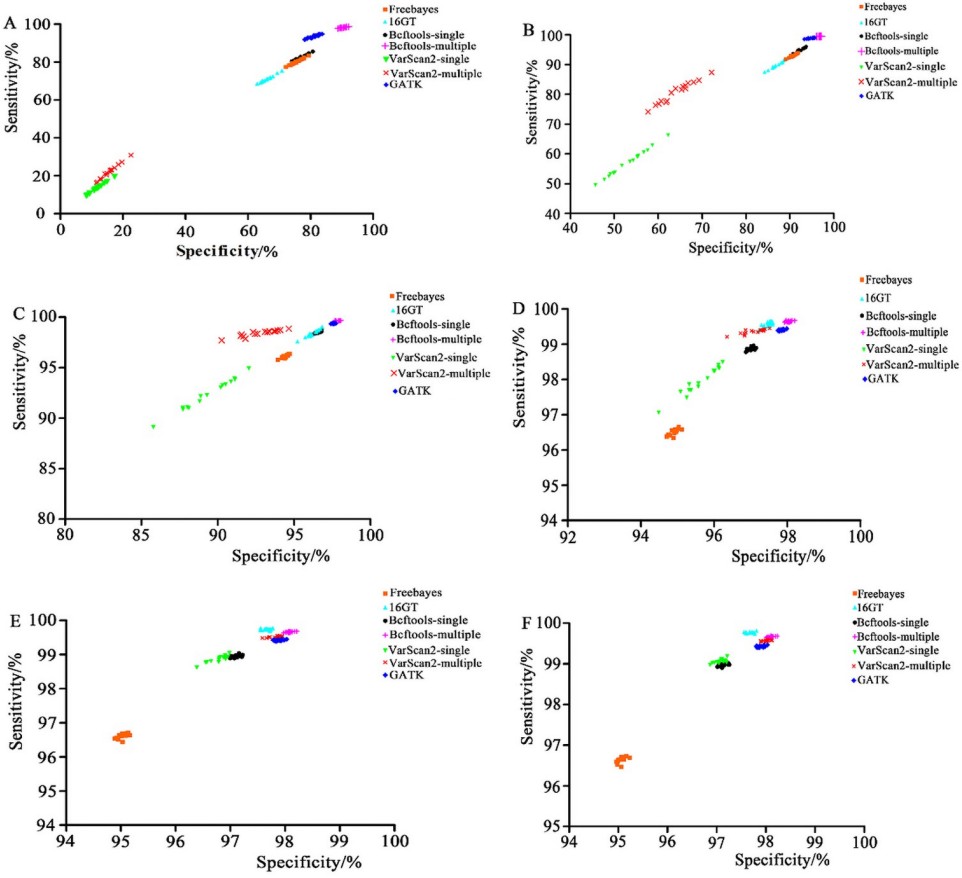

**Fig 4. Two-dimensional scatter plots with specificities and sensitivities of each pipeline at different input read depths.** A, The input read depth is 5X; B, 10X; C, 20X; D, 30X; E, 40X; and F, 50X.

Bcftools-multiple considerably improved the specificity and sensitivity of the detection in comparison with Bcftools-single. For example, under the condition of a 5X input read depth, the specificity increased from 0.771 to 0.905, and the sensitivity increased from 0.827 to 0.982. VarScan2-multiple also improved the performance but not Bcftools-multiple (Fig 5).

## Comparisons of the Ti/Tv ratios of each predictor with different input read depths

The Ti/Tv ratios of each predictor with different input read depths are shown in Fig 6 and S8 Table. From Fig 6, we can see that all Ti/Tv values are between 2.04 and 2.44. No significant ($P < = 0.05$) differences in the ratios were observed among the pipelines with the same input read depths, and among different coverages using the same pipelines in this study. The absolute value of the deviation between the Ti/Tv ratios of the maximum and minimum values in each pipeline did not exceed 0.2, and the absolute deviations of the Ti/Tv ratios of the maximum and minimum values of different pipelines with the same input read depths were less than 0.4 (Fig 6 and S8 Table).

## Discussion

SNPs are widely used in functional gene mapping and population genetics [9,31,32]. As the cost of high-throughput sequencing declined, detecting SNPs from NGS data became

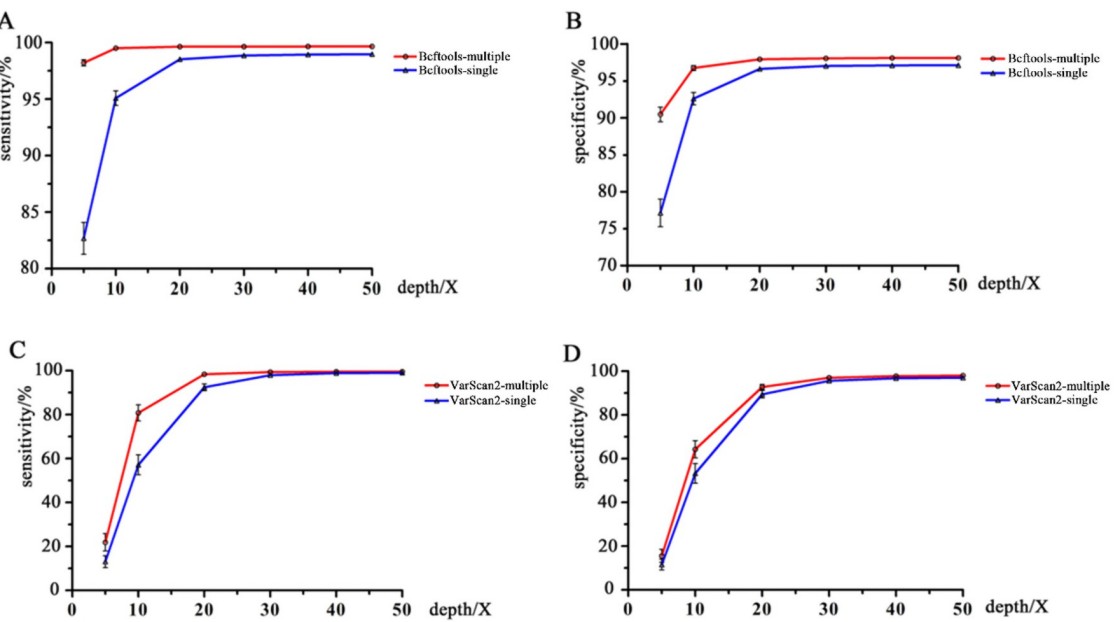

**Fig 5. Comparisons of the sensitivity and specificity of Bcftools and VarScan2 with different sample modes.** A: Comparisons of the sensitivity between Bcftools-single and Bcftools-multiple; B: Comparisons of the specificity between Bcftools-single and Bcftools-multiple; C: Comparisons of the sensitivity between VarScan2-single and VarScan2-multiple; and D: Comparisons of the specificity between VarScan2-single and VarScan2-multiple.

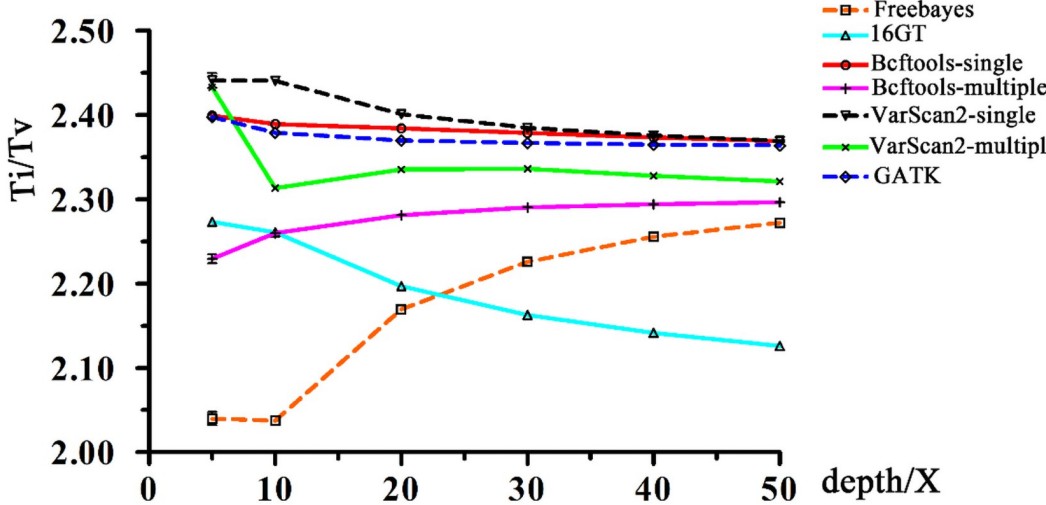

**Fig 6. The transition/transversion ratios of each predictor with different input read depths.**

increasingly common. Generally, NGS data are initially aligned to a reference genome and then subjected to variant calling. Bowtie 2 was chosen to map short reads in the present study since it has a high speed, sensitivity, and accuracy and was particularly good at aligning reads to relatively large genomes (http://bowtie-bio.sourceforge.net/bowtie2/index.shtml) [29]. Many previous studies have reported the capabilities of several available SNP

calling pipelines from NGS data, which were often applied to human data or simulated data [33–36]. GATK is often regarded as the most effective procedure to detect variants from NGS data using resources of known variations, truth sets and other metadata (https://software.broadinstitute.org/gatk/best-practices/about). However, we have fewer known variation resources in poultry than in humans or mice, which may lead to the reduced accuracy of GATK. Ni et al. [7] thought that GATK, SAMtools and Freebayes were all good for processing high-throughput chicken data, but we found that the research in the article used low sequencing depth data, tested relatively few pipelines, and lacked detailed implementation procedures. Thereby, further research was needed. In the present study, we compared the seven SNP calling procedures using 96 NGS datasets with different input read depths of 5X-50X coverage of Rhode Island Red chickens. Luo et al. [19] found that 16GT not only ran fast but also showed the highest sensitivity and specificity in calling SNPs among all tools (GATK UnifedGenotyper, GATK HaplotypeCaller, Freebayes, Fermikit, ISAAC, and VarScan2). In our study, we also found that 16GT was more sensitive than any other pipeline at input read depths ranging from 30X to 50X (Figs 2 and 4), but the specificity of 16GT was moderate (Figs 3 and 4). Freebayes was easy to operate and could be run in one step [18]. However, Freebayes may not be a good pipeline to call SNPs from the short read data sets of the 16 Rhode Island Red chickens due to its unremarkable performances in SNP calling (Figs 1–4). GATK is a popular toolkit and is widely used in many studies [6,37–41]. In our study, the GATK performance was not bad, but at whatever input depth, Bcftools-multiple, and sometimes 16GT, always showed better detection performances than GATK (Figs 1–4). Therefore, we did not recommend GATK for detecting SNPs from chicken NGS data.

A large number of SNPs were detected out by next-generation sequencing, however, we could not evaluate the accuracy of all SNP loci. In order to evaluate the sensitivity and specificity of each SNP calling pipeline, we compared the SNP array genotypes with the genotypes of SNP loci in the array detected by sequencing pipelines with different input read depths, and regarded the array genotyping as the reference data set which were distributed evenly throughout the whole chicken genome. In the present study, 16 chickens were genotyped with the 50K SNP array, and the result was regarded as a standard to evaluate the specificity and sensitivity of each SNP calling pipeline. Since the reference data only consisted of a subset of all SNPs in the genome, the estimated specificity and sensitivity here might differ from the actual values.

The Ti/Tv ratio is also an index used to evaluate the accuracy of SNP calling [40]. A high Ti/Tv ratio (> 2.0) often indicates a high-accuracy SNP set, whereas a low value (~ 0.5) implies low-quality SNP calling [42]. In our study, although each pipeline has a higher or lower value of the Ti/Tv ratio in each different input read depth, all the Ti/Tv ratios fall in the range of 2.04–2.44 (Fig 6, S8 Table), which can be considered as high accurate [42]. Moreover, the Ti/Tv ratio of each pipeline except 16GT approach slowly to around 2.3 with the increase of input read depth (Fig 6, S8 Table), and we speculate that the Ti/Tv = 2.3 could be a genome-wide approximation of chicken in this study.

## Conclusions

In conclusion, (1) if only SNPs were detected, the sequencing depth did not need to exceed 20X since there were no obvious changes in the number of SNPs, sensitivity or specificity beyond 20X. (2) Bcftools-multiple may be the best choice to detect SNPs from chicken NGS data, but for a single sample or a sequencing depth greater than 20X, 16GT was also recommended. Our findings provide a reference for researchers to select suitable pipelines to obtain SNPs from the NGS data of chicken or nonhuman animals.

## Supporting information

**S1 Table. The genotyped results of the Illumina 50 K SNP Beadchip.**
(XLS)

**S2 Table. The sequencing results of 16 Rhode Island Red chickens.**
(XLSX)

**S3 Table. The coverage and alignment rate of each sample.**
(XLSX)

**S4 Table. The total number of SNPs called out by each pipeline in different input depths.**
(XLSX)

**S5 Table. The call rate results of array.**
(XLSX)

**S6 Table. The sensitivity of each pipeline in different input depths.**
(XLSX)

**S7 Table. The specificity of each pipeline in different input depths.**
(XLSX)

**S8 Table. The Ti/Tv ratios of 7 pipelines.**
(XLSX)

**S1 Word. SNP calling pipelines for chicken NGS sets.**
(DOCX)

## Acknowledgments

We wish to thank Wenpeng Han for his help in the experimental methods and polishing of this manuscript during our study.

## Author Contributions

**Conceptualization:** Haigang Bao.

**Data curation:** Jing Liu, Qingmiao Shen, Haigang Bao.

**Formal analysis:** Jing Liu, Haigang Bao.

**Funding acquisition:** Haigang Bao.

**Investigation:** Jing Liu, Haigang Bao.

**Methodology:** Jing Liu, Qingmiao Shen.

**Software:** Jing Liu, Haigang Bao.

**Supervision:** Haigang Bao.

**Validation:** Jing Liu, Haigang Bao.

**Visualization:** Jing Liu, Haigang Bao.

**Writing – original draft:** Jing Liu, Qingmiao Shen.

**Writing – review & editing:** Jing Liu, Haigang Bao.

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
