## [Decision Letter · Decision Letter 0]

18 Jun 2021

PONE-D-21-09642

Comparison of seven SNP calling pipelines for the next-generation sequencing data of chickens

PLOS ONE

Dear Dr. Bao,

Thank you for submitting your manuscript to PLOS ONE. After careful consideration, we feel that it has merit but does not fully meet PLOS ONE’s publication criteria as it currently stands. Therefore, we invite you to submit a revised version of the manuscript that addresses the points raised during the review process.

You are required to revise according to the comments from the reviewer with major points addressed properly and minor points recommended to change if appropriate.

We look forward to receiving your revised manuscript.

Kind regards,

Shu-Biao Wu, PhD

Academic Editor

PLOS ONE

2. Thank you for stating the following in the manuscript:

This study was supported by the Modern Agricultural Industry Technology System of China [grant number CARS-40]. The funder did not play any role in the design of the study, collection, analysis, interpretation of data or writing the manuscript.

However, funding information should not appear in the areas of your manuscript. We will only publish funding information present in the Funding Statement section of the online submission form.

Thank you for stating the following in the manuscript:

This study was supported by the Modern Agricultural Industry Technology System of China [grant number CARS-40]. The funder did not play any role in the design of the study, collection, analysis, interpretation of data or writing the manuscript.

Please include your amended statements within your cover letter; we will change the online submission form on your behalf."

Additional Editor Comments (if provided):

Reviewers' comments:

Reviewer's Responses to Questions

**Comments to the Author**

1. Is the manuscript technically sound, and do the data support the conclusions?

Reviewer #1: Yes

2. Has the statistical analysis been performed appropriately and rigorously? 

Reviewer #1: Yes

3. Have the authors made all data underlying the findings in their manuscript fully available?

Reviewer #1: Yes

4. Is the manuscript presented in an intelligible fashion and written in standard English?

Reviewer #1: Yes

5. Review Comments to the Author

Reviewer #1: GENERAL COMMENTS:

As well as the rapid advance of NGS techniques for large-scale production of genomic data, many research groups also quickly advanced in the development of pipelines to deal with this type of data. Some pipelines have become gold standards, but not necessarily because they are the best, but the most used and cited. This study is very relevant to help bioinformats to apply the proper method to identify SNPs using high-throughput sequencing technologies. In addition, the article explores different scenarios faced by researchers regarding sequencing coverage, which is often limited due to the researcher's resources available. Although the study was performed using the chicken model, it can be easily extrapolated to any other species.

Overall, I really enjoyed the study in all aspects, but I was a little confused in the "Analysis of the sensitivity and specificity of SNP-calling pipelines" part of which I recommend a major review.

Major:

Line 143 – In my understanding you have settled 4 categories to the SNPs to validate it when comparing the SNP panel with your “sequencing data” right? However, it is not clear to me (in your writing) which set of your sequencing data you have used. According to your data and results I see that you have used the data from the 16 individuals separately according to the depth of coverage (as like all the other comparisons). And in addition, you have also compared according to the SNP caller tools. If so, you must add this information clearly in your methods, in your results, and also take it in consideration when discussing.

Minor:

ABSTRACT

Line 28 – please, replace ”object” to objective, or goal, or aim.

Line 27-29 - This phrase strikes me as a bit “scientifically selfish”. You are making it public to allow other scientists to use it, right? I would like to suggest something more general like this: “We took advantage of a gap knowledge in selecting the appropriated SNP calling pipeline to handle with high-throughput NGS data. To fill this gap., we studied and compared seven SNP calling pipelines, which include… and also using the different coverage deph…”

Introduction

Line 95 – replace traits to “advantages and disadvantages”, or something like this, otherwise, the sentence does not say that much.

SECTION “NGS DATA SETS AND SNP CALLING PIPELINES.

Line 133 – Please, describe better the quality control. Please, replace “clean”, to cleaned (after quality control, right?)

From lines 138 to 142 – Have you used “default” parameters for all the pipelines described here? You should better explain it on the manuscript body or describe the parameters used. In your supplementary file you say “All results of this study depended on the default parameters used in each pipeline. Any change of these parameters may alter results and conclusions.”. Looks like you have defined the “default” as the parameters you have used in your pipeline, but what about the “program’s parameters”? Have you used it? You need a very briefly sentence saying that you have used: or “all the program´s default” or modified defaults according to the supplementary file. However, if you will modify de program´s default, you should also briefly justify it.

RESULTS

172-174 – Please, clarify this sentence!

217-218 – replace “discovered” to “observed”

Figure 1: I suggest you replace A and B. So, you will have all the figures standardize with the same “artistic” style. But it is just a suggestion.

265 – Please, you need to describe it better.

Discussion

Line 279 – Here you are saying that you have used the program´s default parameters… So please, be concise with your results.

Line 280-281- I do not consider this information relevant in the way is written. I would like to suggest to you write that you chose not to change the parameters to represent a scenario commonly used by researchers, however, any change to these parameters needs to be carefully done and properly justified. Because this is the reality. The algorithm defaults are usually settled by researchers of exact sciences and must be altered when biologically they do not make sense.

Line 283 – I think the correct term here is “large” genomes, please check it.

Line 308 – Please, I think you can improve the writing here. Something like this: Moreover, GATK was more time consuming than…or the most time consuming…

In addition, I do not remember seeing any mention in your results about processing time. If you have this information, please add it to your table 1 and write a sentence about it there (more than just the features column). Otherwise, you cannot argue based on your pipeline, you need to define that this information comes from other references. Moreover, when you write about “time consuming”, you should stablish a several of other standardized criteria, like number of cores, memory used, etc and etc, for each one of the approaches you have worked with…

Line 310 – “It was not possible” sounds better.

From 310 and 311 – I did not get the point of the first and second sentence, how they connect with each other? It was not possible because polymorphic SNP loci formed good sampling data for all SNPs? Please clarify this.

Moreover, you cannot base your discussion in your opinion. How does the evidence from this study support its conclusion?

Line 310 to 317. Please, reorganize this whole paragraph.

Line 318 to 319 – Please, add the “high ratio” value same as you did to the low. In addition, use > and < if possible.

Line 320-322 – Please, you should be more straight forward and explore better this last paragraph using your results. First of all, these “validation” using the SNP panel was performed comparing with your 16 individuals unregard to the X coverage? No, I see you have compared all the possible scenarios. Please, explore it. Look:

In our study, although pipeline X has a higher or lower ratio or etc, etc., all the xxx ratios fall in the range of XXXX, which can be considered as “??””( High accurate?). Moreover, no significant (Pvalue<=???) differences in the ratios were observed among the pipelines used in this study??? And about among different coverages??

Here you really must point to the readers that although all pipelines have defined good accuracy by the literature...your study indicates that in some situations you can have a better accuracy (is that the case?). This accuracy is dependent of the coverage? the used pipeline? Please, explore it better.

6. PLOS authors have the option to publish the peer review history of their article (what does this mean?). If published, this will include your full peer review and any attached files.

Reviewer #1: **Yes: **Fábio Pértille

---

## [Author Response · Author response to Decision Letter 0]

22 Jul 2021

Responses to reviewers’ comments

Reviewer #1: GENERAL COMMENTS:

As well as the rapid advance of NGS techniques for large-scale production of genomic data, many research groups also quickly advanced in the development of pipelines to deal with this type of data. Some pipelines have become gold standards, but not necessarily because they are the best, but the most used and cited. This study is very relevant to help bioinformats to apply the proper method to identify SNPs using high-throughput sequencing technologies. In addition, the article explores different scenarios faced by researchers regarding sequencing coverage, which is often limited due to the researcher's resources available. Although the study was performed using the chicken model, it can be easily extrapolated to any other species.

Overall, I really enjoyed the study in all aspects, but I was a little confused in the "Analysis of the sensitivity and specificity of SNP-calling pipelines" part of which I recommend a major review.

Major:

Line 143 – In my understanding you have settled 4 categories to the SNPs to validate it when comparing the SNP panel with your “sequencing data” right? However, it is not clear to me (in your writing) which set of your sequencing data you have used. According to your data and results I see that you have used the data from the 16 individuals separately according to the depth of coverage (as like all the other comparisons). And in addition, you have also compared according to the SNP caller tools. If so, you must add this information clearly in your methods, in your results, and also take it in consideration when discussing.

Response:

We add the information clearly in the methods, results, and discussion as follows:

Line 144: add “We compared the SNP array genotypes with the genotypes of SNP loci in the array detected by sequencing pipelines. In order to assess the sensitivity, and specificity of the pipelines with input read depth gradients of 5X-50X coverage,” at the beginning of the paragraph.

Line 199: add “To assess the sensitivity, and specificity of each pipeline with different input read depths,” at the beginning of the paragraph.

Line 201: replace “, and its results” with “. We compared the SNP array genotypes with the genotypes of SNP loci in the array detected by sequencing pipelines, and the array results”.

Line 295: insert “with different input read depths of 5X-50X coverage” between “datasets” and “of Rhode”.

Line 310 -313: replace “It was not possible for us to evaluate the detection accuracy of all SNP loci. The SNPs in the 50K SNP array were distributed evenly throughout the whole chicken genome. In our opinion, polymorphic SNP loci detected using the 50K SNP array formed good sampling data for all SNPs in the chicken genome.” with “A large number of SNPs were detected out by next-generation sequencing, however, we could not evaluate the accuracy of all SNP loci. In order to evaluate the sensitivity and specificity of each SNP calling pipeline, we compared the SNP array genotypes with the genotypes of SNP loci in the array detected by sequencing pipelines with different input read depths, and regarded the array genotyping as the reference data set which were distributed evenly throughout the whole chicken genome.”.

Minor:

ABSTRACT

Line 28 – please, replace ”object” to objective, or goal, or aim.

Response:

Line 28: We replace the sentence as following: Line 27 - 33.

Line 27-29 - This phrase strikes me as a bit “scientifically selfish”. You are making it public to allow other scientists to use it, right? I would like to suggest something more general like this: “We took advantage of a gap knowledge in selecting the appropriated SNP calling pipeline to handle with high-throughput NGS data. To fill this gap., we studied and compared seven SNP calling pipelines, which include… and also using the different coverage depth…”

Response:

Line 27 -33: replace the sentence “Our object was to select a high-performance SNP calling pipeline for chicken NGS data for application in our future studies. Here, we studied the performances of seven SNP calling pipelines, including the 16GT, genome analysis toolkit (GATK), Bcftools-single (Bcftools single sample mode), Bcftools-multiple (Bcftools multiple sample mode), VarScan2-single (VarScan2 single sample mode), VarScan2-multiple (VarScan2 multiple sample mode) and Freebayes pipelines, using 96 NGS data from 16 Rhode Island Red chickens.” with “We took advantage of a gap knowledge in selecting the appropriated SNP calling pipeline to handle with high-throughput NGS data. To fill this gap, we studied and compared seven SNP calling pipelines, which include 16GT, genome analysis toolkit (GATK), Bcftools-single (Bcftools single sample mode), Bcftools-multiple (Bcftools multiple sample mode), VarScan2-single (VarScan2 single sample mode), VarScan2-multiple (VarScan2 multiple sample mode) and Freebayes pipelines, using 96 NGS data with the different depth gradients of approximately 5X, 10X, 20X, 30X, 40X, and 50X coverage from 16 Rhode Island Red chickens.”.

Introduction

Line 95 – replace traits to “advantages and disadvantages”, or something like this, otherwise, the sentence does not say that much.

Response:

Line 95: replace “traits” with “advantages and disadvantages”

SECTION “NGS DATA SETS AND SNP CALLING PIPELINES.

Line 133 – Please, describe better the quality control. Please, replace “clean”, to cleaned (after quality control, right?)

Response:

Line 133: add a new sentence “Cleaned reads were obtained by Trimmomatic (version 0.39; S1 Word) from raw sequencing data.” before “After quality control”.

S1 Word: add a new paragraph at the beginning of this supplementary file as follows:

“1. Qualitative Control with Trimmomatic (version 0.39)

java -jar trimmomatic-0.39.jar PE -threads 16 Sample_1.clean.fq Sample_2.clean.fq Sample_forward_paired.fq Sample_forward_unpaired.fq Sample_reverse_paired.fq Sample_reverse_unpaired.fq ILLUMINACLIP:TruSeq3-PE-2.fa:2:30:10 LEADING:3 TRAILING:3 SLIDINGWINDOW:4:15 CROP:135 MINLEN:135” .

Line 133, Line 168 and L170: replace “clean data” with “cleaned data”.

From lines 138 to 142 – Have you used “default” parameters for all the pipelines described here? You should better explain it on the manuscript body or describe the parameters used. In your supplementary file you say “All results of this study depended on the default parameters used in each pipeline. Any change of these parameters may alter results and conclusions.”. Looks like you have defined the “default” as the parameters you have used in your pipeline, but what about the “program’s parameters”? Have you used it? You need a very briefly sentence saying that you have used: or “all the program´s default” or modified defaults according to the supplementary file. However, if you will modify de program´s default, you should also briefly justify it.

Response:

Line 141: add a new sentence “All results of this study depended on programs’ defaults in each pipeline.” before “Details of processing”.

S1 Word: delete the first two sentences “All results of this study depended on the default parameters used in each pipeline. Any change of these parameters may alter results and conclusions.”.

RESULTS

172-174 – Please, clarify this sentence!

Response:

Line 172-174: replace “Bowtie 2 mapped short reads to the chicken reference genome (Gallus_gallus-5.0), and the average mapping ratios were approximately 94% (S3 Table).” with “Paired-end cleaned reads were aligned against the chicken reference genome (Gallus_gallus-5.0) using Bowtie 2 (version 2.2.9). A summary of cleaned data alignments is displayed in S3 Table. The alignment rate of the cleaned data of each sample was between 90.91% and 95.21% (S3 Table).”.

217-218 – replace “discovered” to “observed”

Response:

Line 217 -218: replace “discovered” with “observed”

Figure 1: I suggest you replace A and B. So, you will have all the figures standardize with the same “artistic” style. But it is just a suggestion.

Response:

Figure 1: We have replaced A and B.

265 – Please, you need to describe it better.

Response:

Line 265: replace “Comparisons of the Ti/Tv of SNPs detected by different SNP calling pipelines” with “Comparisons of the Ti/Tv ratios of each predictor with different input read depths”

Discussion

Line 279 – Here you are saying that you have used the program´s default parameters… So please, be concise with your results.

Line 280-281- I do not consider this information relevant in the way is written. I would like to suggest to you write that you chose not to change the parameters to represent a scenario commonly used by researchers, however, any change to these parameters needs to be carefully done and properly justified. Because this is the reality. The algorithm defaults are usually settled by researchers of exact sciences and must be altered when biologically they do not make sense.

Response:

Line 141: add a new sentence “All results of this study depended on programs’ defaults in each pipeline.” before “Details of processing”.

Line 279-281: delete the sentence “Throughout this study, the default parameters were used in each pipeline. Any change in these parameters may lead to different results, and another pipeline might perform better under different parameter settings.”. 

Line 283 – I think the correct term here is “large” genomes, please check it.

Response:

Line 283: replace “long” with “large”.

Line 308 – Please, I think you can improve the writing here. Something like this: Moreover, GATK was more time consuming than…or the most time consuming…

In addition, I do not remember seeing any mention in your results about processing time. If you have this information, please add it to your table 1 and write a sentence about it there (more than just the features column). Otherwise, you cannot argue based on your pipeline, you need to define that this information comes from other references. Moreover, when you write about “time consuming”, you should stablish a several of other standardized criteria, like number of cores, memory used, etc and etc, for each one of the approaches you have worked with…

Response:

Line 299: delete “a more time-saving pipeline (data not shown) and”.

Line 307-308: delete the sentence “Moreover, GATK often spent a long time (data not shown) performing its complicated implementation steps.”

Line 310 – “It was not possible” sounds better.

From 310 and 311 – I did not get the point of the first and second sentence, how they connect with each other? It was not possible because polymorphic SNP loci formed good sampling data for all SNPs? Please clarify this. Moreover, you cannot base your discussion in your opinion. How does the evidence from this study support its conclusion? Line 310 to 317. Please, reorganize this whole paragraph.

Response:

Line 310 -313: replace the sentences “It was not possible for us to evaluate the detection accuracy of all SNP loci. The SNPs in the 50K SNP array were distributed evenly throughout the whole chicken genome. In our opinion, polymorphic SNP loci detected using the 50K SNP array formed good sampling data for all SNPs in the chicken genome.” with “A large number of SNPs were detected out by next-generation sequencing, however, we could not evaluate the accuracy of all SNP loci. In order to evaluate the sensitivity and specificity of each SNP calling pipeline, we compared the SNP array genotypes with the genotypes of SNP loci in the array detected by sequencing pipelines with different input read depths, and regarded the array genotyping as the reference data set which were distributed evenly throughout the whole chicken genome.”.

Line 318 to 319 – Please, add the “high ratio” value same as you did to the low. In addition, use > and < if possible.

Response:

Line 318-319: change “A high Ti/Tv ratio” to “A high Ti/Tv ratio (> 2.0)”

Line 320-322 – Please, you should be more straight forward and explore better this last paragraph using your results. First of all, these “validation” using the SNP panel was performed comparing with your 16 individuals unregard to the X coverage? No, I see you have compared all the possible scenarios. Please, explore it. Look:

In our study, although pipeline X has a higher or lower ratio or etc, etc., all the xxx ratios fall in the range of XXXX, which can be considered as “??””( High accurate?). Moreover, no significant (Pvalue<=???) differences in the ratios were observed among the pipelines used in this study??? And about among different coverages??

Here you really must point to the readers that although all pipelines have defined good accuracy by the literature...your study indicates that in some situations you can have a better accuracy (is that the case?). This accuracy is dependent of the coverage? the used pipeline? Please, explore it better.

Response:

In our study, we calculated the Ti/Tv ratios of each pipeline in different input read depths, however, all Ti/Tv values are between 2.04 and 2.44. The expected Ti/Tv ratios in whole-genome sequencing are 2.10 and 2.07 for known and novel variants, respectively in human as Liu et al. reported, but it has not been reported in chickens. The Ti/Tv ratios of seven SNP calling pipelines in our study are greater than 2.0 with no significance. All seven SNP calling pipelines perform well in this index and the Ti/Tv ratios does not account for the excellence of each tool. 

Line 267：insert a new sentence “No significant (P <=0.05) differences in the ratios were observed among the pipelines with the same input read depths, and among different coverages using the same pipelines in this study.” between “2.44.” and “The”.

Line 320-322: replace the sentence “In the present study, all Ti/Tv ratios fall in the range of 2.04-2.44 (Fig 6, S8 Table), and we cannot conclude from these data that there are significant differences in the SNP accuracy of different pipelines” with “In our study, although each pipeline has a higher or lower value of the Ti/Tv ratio in each different input read depth, all the Ti/Tv ratios fall in the range of 2.04-2.44 (Fig 6, S8 Table), which can be considered as high accurate [42]. Moreover, the Ti/Tv ratio of each pipeline except 16GT approach slowly to around 2.3 with the increase of input read depth (Fig 6, S8 Table), and we speculate that the Ti/Tv = 2.3 could be a genome-wide approximation of chicken in this study.”

In addition, according to editor’s comments, we also made the following changes:

Line 5: delete “Wenpeng Han,2”

Line 9: delete “2 Beijing Huadu Yukou Poultry Industry Co. Ltd., Beijing 101206, China”

Line 12-13: replace “*Corresponding author: zjbhg@126.com Telephone number: +86-10-62734828” with “*Corresponding author E-mail: zjbhg@126.com (HB)”.

Line 14-15: delete “Address: Room 437, Animal Science Building, NO.2 Yuanmingyuan West Road, Haidian District, Beijing 100193, China”

Line 363-366: delete “Funding This study was supported by the Modern Agricultural Industry Technology System of China [grant number CARS-40]. The funder did not play any role in the design of the study, collection, analysis, interpretation of data or writing the manuscript.”

Line 371, 373, 375 and 376: delete “, Wenpeng Han”

Line 378: replace “Not applicable” with “We wish to thank Wenpeng Han for his support in our work and valuable feedback on this manuscript”

---

## [Decision Letter · Decision Letter 1]

30 Dec 2021

Comparison of seven SNP calling pipelines for the next-generation sequencing data of chickens

PONE-D-21-09642R1

Dear Dr. Bao,

We’re pleased to inform you that your manuscript has been judged scientifically suitable for publication and will be formally accepted for publication once it meets all outstanding technical requirements.

Kind regards,

Shu-Biao Wu, PhD

Academic Editor

PLOS ONE

Additional Editor Comments (optional):

Reviewers' comments:

Reviewer's Responses to Questions

**Comments to the Author**

1. If the authors have adequately addressed your comments raised in a previous round of review and you feel that this manuscript is now acceptable for publication, you may indicate that here to bypass the “Comments to the Author” section, enter your conflict of interest statement in the “Confidential to Editor” section, and submit your "Accept" recommendation.

Reviewer #1: All comments have been addressed

2. Is the manuscript technically sound, and do the data support the conclusions?

Reviewer #1: Yes

3. Has the statistical analysis been performed appropriately and rigorously? 

Reviewer #1: Yes

4. Have the authors made all data underlying the findings in their manuscript fully available?

Reviewer #1: (No Response)

5. Is the manuscript presented in an intelligible fashion and written in standard English?

Reviewer #1: (No Response)

6. Review Comments to the Author

Reviewer #1: The authors adequately answered all my questions and responded to all my suggestions. my only concern is where the data may be found? Please, provide an URL or more details, I could not find/access the provided DB.

In addition, I would like to add that my role as reviewer is not to ensure that the grammar or language style is impeccable.

7. PLOS authors have the option to publish the peer review history of their article (what does this mean?). If published, this will include your full peer review and any attached files.

Reviewer #1: No

---

## [Editor Report · Acceptance letter]

21 Jan 2022

PONE-D-21-09642R1 

Comparison of seven SNP calling pipelines for the next-generation sequencing data of chickens 

Dear Dr. Bao:

I'm pleased to inform you that your manuscript has been deemed suitable for publication in PLOS ONE. Congratulations! Your manuscript is now with our production department. 

Kind regards, 

on behalf of

Dr. Shu-Biao Wu 

Academic Editor

PLOS ONE